# Rethink GraphODE Generalization within Coupled Dynamical System

Guancheng Wan [1]  Zijie Huang [1]  Wanjia Zhao [2]  Xiao Luo [1]  Yizhou Sun [1]  Wei Wang [1]

## Abstract

Coupled dynamical systems govern essential phenomena across physics, biology, and engineering, where components interact through complex dependencies. While Graph Ordinary Differential Equations (GraphODE) offer a powerful framework to model these systems, their **generalization** capabilities degrade severely under limited observational training data due to two fundamental flaws: (i) the entanglement of static attributes and dynamic states in the initialization process, and (ii) the reliance on context-specific coupling patterns during training, which hinders performance in unseen scenarios. In this paper, we propose a Generalizable GraphODE with disentanglement and regularization (`GREAT`) to address these challenges. Through systematic analysis via the Structural Causal Model, we identify backdoor paths that undermine generalization and design two key modules to mitigate their effects. The *Dynamic-Static Equilibrium Decoupler (DyStaED)* disentangles static and dynamic states via orthogonal subspace projections, ensuring robust initialization. Furthermore, the *Causal Mediation for Coupled Dynamics (CMCD)* employs variational inference to estimate latent causal factors, reducing spurious correlations and enhancing universal coupling dynamics. Extensive experiments across diverse dynamical systems demonstrate that ours outperforms state-of-the-art methods within both in-distribution and out-of-distribution.

## 1. Introduction

Dynamical systems describe the evolution of states over time and are fundamental to a wide range of scientific and engineering disciplines (Kipf et al., 2018; Huang et al., 2021; Khan, 2013). A prominent subset of these systems is

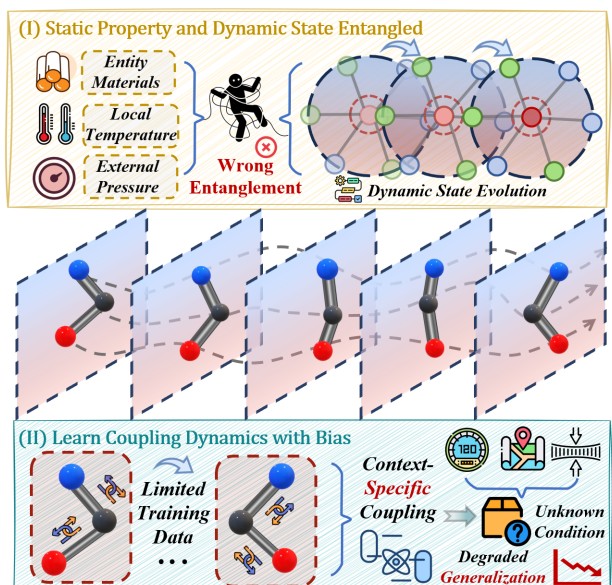

*Figure 1.* **Problem Illustration.** In coupled dynamical systems, GraphODE models rely on partially observed trajectories, which can result in a bias towards the training data distribution. Specifically, **I)** static attributes such as material properties or environmental conditions influence the initialization of system states but should not affect the dynamic evolution governed by the ODE function. **II)** Context-specific coupling during training often leads to biased interaction patterns, which hinders the ability of the model to generalize across different distributions and conditions.

**coupled dynamical systems** (Bahsoun & Liverani, 2024; Börner et al., 2024; Sun & Yu, 2017), in which multiple interconnected components interact to produce complex collective behaviors. Such systems are ubiquitous across various fields, including physics, biology, and engineering (Gambuzza et al., 2020; Wu et al., 2024b). For example, in electrical engineering, coupled oscillators are employed to model circuits where components like inductors and capacitors interact, giving rise to phenomena such as synchronization and resonance (Galias, 2019). Accurately modeling these interactions is crucial for understanding and predicting the behavior of complex systems. Traditional numerical methods often necessitate substantial domain-specific expertise and computational resources, which can limit their scalability to more intricate systems (de Arruda & Moreno, 2024). As a result, data-driven approaches have emerged as effective alternatives for capturing the dynamics of these

[1]University of California, Los Angeles [2]Stanford University. Correspondence to: Guancheng Wan <gcwan03@ucla.edu>.

*Proceedings of the 42nd International Conference on Machine Learning*, Vancouver, Canada. PMLR 267, 2025. Copyright 2025 by the author(s).

systems (Rubanova et al., 2019; Greydanus et al., 2019).

Among these approaches, graphs provide an effective means to model the intricate interactions by representing entities as nodes and their interactions as edges. Graph Neural Networks (GNNs) (Kipf & Welling, 2017; Veličković et al., 2017) leverage this structure to learn complex dependencies among components, capturing both individual behaviors and their interrelations. To further model the continuous evolution of such systems, the Graph Ordinary Differential Equation (GraphODE) has been introduced (Luo et al., 2023; Qin et al., 2024), integrating GNNs with differential equation solvers to seamlessly capture temporal dynamics. However, these models often rely on partially observed trajectories during training, which can introduce bias toward the training data distribution (Huang et al., 2024a; 2023b). Consequently, when deployed in unseen environments, these models may struggle to capture universal physical laws, leading to reduced **generalization ability**.

Building on this discussion, the development of a **Generalizable GraphODE**, capable of modeling complex system dynamics across diverse environments, becomes critical. Existing GraphODE architectures generally consist of three components: an encoder, a processor, and a decoder (Huang et al., 2020). The encoder initializes the system's state, which is subsequently processed to model its temporal evolution. However, previous work neglects that this initial state often encapsulates various factors, including certain **static attributes** that are unrelated to the **dynamic evolution** governed by the ODE function. For example, in a multi-pendulum system, attributes such as material composition or local temperature represent "static states" with minimal influence on the pendulum's dynamic motion. Suppose the differential process inadvertently incorporates the effects of these static factors into the dynamic evolution. In that case, it can negatively impact the model's generalization capabilities when applied to new environments or varying static properties. This observation raises a critical question: **I)** *How can we disentangle static attributes from dynamic states in the initialization process?*

Prior research has predominantly focused on refining the encoder to improve the initialization of system states. However, it has largely overlooked the crucial role of the processor, which is responsible for learning the GraphODE function that governs the system's evolution. In the context of coupled interactions, we introduce the **coupling factor**, a control variable that influences the interaction between two entities. For example, in a multi-pendulum system, this factor quantifies the strength of the coupling force exerted by a spring on each pendulum. During the training phase, however, this coupling factor often captures context-specific interaction patterns present in the training data. This leads to a dynamic process that is overly dependent on the

training distribution, potentially compromising the model's performance in novel scenarios or under different initial conditions (*e.g.*, velocity or location). This raises another question: **II)** *How can we design a processor that learns universal coupling dynamics without bias?*

To simultaneously address the challenges mentioned above, in this paper, we propose **G**eneralizable **GR**aphODE with dis**E**ntanglement **A**nd regulariza**T**ion (GREAT) framework. Through the lens of Structural Causal Models (SCM), as detailed in Sec. 2.2, we identify two potential backdoor paths that hinder the generalization abilities of GraphODE, corresponding to the issues outlined earlier. Therefore, based on the SCM, to answer question **I)**, we propose the *Dynamic-Static Equilibrium Decoupler (DyStaED)*. At the encoder level, this mechanism disentangles static and dynamic states through two distinct subspace projections constrained by orthogonality. This separation ensures that static attributes, such as material properties or ambient conditions, do not interfere with the dynamic evolution process. Furthermore, we leverage the self-exciting nature of the Hawkes process to augment dynamic states, thereby enhancing temporal dependencies and preparing robust initializations for dynamic evolution. To address **II)**, we introduce *Causal Mediation for Coupled Dynamics (CMCD)*, which mitigates spurious correlations between the coupling factor and contextual states. By utilizing variational inference, we estimate unobserved factors that influence coupling dynamics. This estimation enables us to regularize causal relationships within the state evolution, ensuring that the learned dynamics are not biased by specific training data distributions. Our principal contributions are summarized as follows:

❶ *Problem Identification.* We are the first to apply Structural Causal Models to systematically investigate Generalizable GraphODE. We identify two potential backdoor paths that hinder generalization and propose corresponding design principles to address these challenges.

❷ *Practical Solution.* We introduce the Dynamic-Static Equilibrium Decoupler, which disentangles static attributes from dynamic states through orthogonal subspace projections. Additionally, we propose Causal Mediation, which uses variational inference to estimate latent causal factors, mitigating spurious correlations and enhancing the universal coupling dynamics.

❸ *Experimental Validation.* We conduct comprehensive experiments on multiple benchmarks involving diverse coupled dynamical systems, demonstrating that GREAT significantly outperforms state-of-the-art methods in both in-distribution and out-of-distribution settings.

## 2. Motivation

### 2.1. Preliminaries

**Notations.** A coupled dynamical system is represented as a temporal graph $\mathcal{G} = (\mathcal{V}, \mathcal{E}, \mathcal{X})$, where $\mathcal{V}$ is the set of $N$

entities (nodes), $\mathcal{E} \subseteq \mathcal{V} \times \mathcal{V}$ denotes the edges describing interactions between entities, and $\mathcal{X}$ is the feature matrix that records the entity states over time. Each entity $i$ at timestamp $t$ has a feature vector $\mathbf{x}_i(t) \in \mathbb{R}^d$, where $d$ is the feature dimension. The adjacency matrix $\mathbf{A} \in \mathbb{R}^{N \times N}$ encodes the pairwise relationships, where $\mathbf{A}_{uv} = 1$ if an edge $e_{uv} \in \mathcal{E}$ exists at time $t$, otherwise $\mathbf{A}_{uv} = 0$. The diagonal degree matrix $\mathbf{D}$ is defined as $\mathbf{D}_{uu} = \sum_v \mathbf{A}_{uv}$, and the normalized adjacency matrix is given by $\hat{\mathbf{A}} = \mathbf{D}^{-1/2} \mathbf{A} \mathbf{D}^{-1/2}$. The system's dynamics can further be described using the Laplacian matrix $\mathbf{L} = \mathbf{D} - \mathbf{A}$ or its symmetric normalized form $\tilde{\mathbf{L}} = \mathbf{I} - \hat{\mathbf{A}}$, where $\mathbf{I}$ is the identity matrix.

**Neural ODEs for Dynamical Systems.** The continuous evolution of entity states in a coupled dynamical system can be modeled using Neural Ordinary Differential Equations (Neural ODEs) (Chen et al., 2018; Alvarez et al., 2020). Each entity $i \in \mathcal{V}$ has a latent state $h_i(t) \in \mathbb{R}^h$, where $h$ represents the latent dimension. The state evolution over time is governed by the following ODE:

$$\frac{dh_i(t)}{dt} = g_\theta \left( h_i(t), \{h_j(t) : j \in \mathcal{N}(i)\}, \mathbf{A} \right),$$
$$h_i(t_0) = f_{\text{enc}} \left( \mathbf{X}(t_{-M} : t_{-1}), \mathcal{G} \right), \quad (1)$$

where $g_\theta$ is a learnable neural network parameterized by $\theta$, which models the dynamics of entity $i$ based on its own latent state $h_i(t)$ and the aggregated influence of its neighbors $\mathcal{N}(i)$ using the graph structure $\mathbf{A}$. The encoder $f_{\text{enc}}$ initializes the latent states $h_i(t_0)$ by mapping historical states $\mathbf{X}(t)$ and graph information into the latent space. The latent state $h_i(t)$ at any future time $t$ is obtained by integrating the ODE with respect to time $t$, typically solved using numerical solvers such as Euler's method or Runge-Kutta methods:

$$h_i(t) = h_i(t_0) + \int_{t_0}^{t} g_\theta \left( h_i(\tau), \{h_j(\tau) : j \in \mathcal{N}(i)\}, \mathbf{A} \right) d\tau.$$
$$(2)$$

The predicted entity states $\hat{\mathbf{x}}_i(t)$ are reconstructed from the latent states $h_i(t)$ through a decoder $f_{\text{dec}}$: $\hat{\mathbf{x}}_i(t) = f_{\text{dec}} \left( h_i(t) \right)$. The objective is to minimize the difference between the predicted trajectories $\hat{\mathbf{X}}(t)$ and the ground truth $\mathbf{X}(t)$ over the prediction horizon $t \in [t_0, T]$. The loss function is expressed as:

$$\mathcal{L}_{\text{predict}} = \sum_{i=1}^{N} \sum_{k=1}^{K} \left\| \mathbf{X}_i(t_k) - \hat{\mathbf{X}}_i(t_k) \right\|_2^2, \quad (3)$$

where $t_k$ represents the timestamp.

## 2.2. A Causal View on Generalizable GraphODE

**Definition 2.1** (Coupled Dynamical System). *Coupled dynamical system refers to a system where the state evolution of one entity is inherently influenced by its interactions with others, often leading to energy transfer or synchronization. The interactions are governed by physics-based coupling mechanisms that determine energy/information exchange.*

For instance, consider two pendulums connected by a

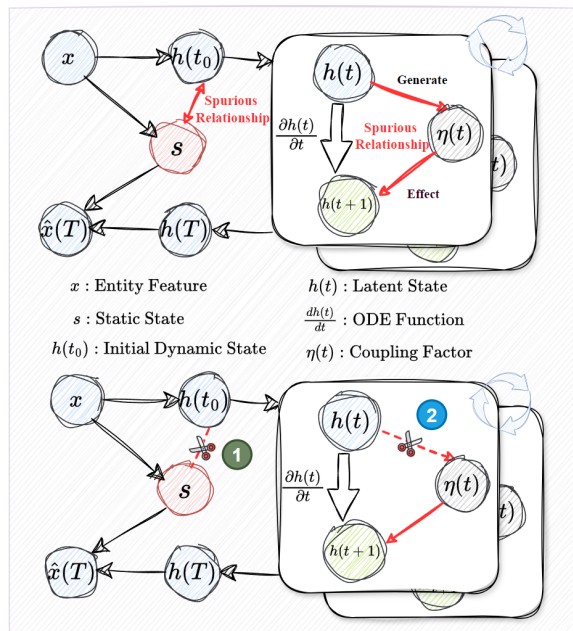

*Figure 2.* Structural Causal Model (SCM) for Generalizable GraphODE within Coupled Systems.

spring—a canonical example of coupled dynamics. Their motion can be described by: $m\ddot{x} = -mg\frac{x}{l_1} - k(x - y)$, $m\ddot{y} = -mg\frac{y}{l_2} + k(x-y)$, where the spring constant $k$ embodies the coupling mechanism. Similar principles govern interactions in molecular systems (harmonic oscillators) and electrical circuits (coupled RLC networks), demonstrating the universality of coupling phenomena across scales. Effectively modeling these systems demands frameworks that reconcile physics-based interactions with nonlinear heterogeneous dynamics in complex environments, motivating our generalizable GraphODE:

**Definition 2.2** (Generalizable GraphODE for Coupled Systems). *To describe the dynamics of coupled systems, we define a **Generalizable GraphODE** as:*

$$\frac{\partial h_i(t)}{\partial t} = \sum_{j \in \mathcal{N}(i)} \eta_{i,j}(t) \cdot \Phi(h_j(t), h_i(t); \mathbf{\Theta}_{\text{ode}}), \quad (4)$$

*where $h_i(t) \in \mathbb{R}^h$ represents the latent state of entity $i$ at time $t$, and $\eta_{i,j}(t)$ is the **coupling factor** quantifying the interaction strength between entities $i$ and $j$. The interaction function $\Phi(h_j(t), h_i(t); \mathbf{\Theta}_{ode})$ models nonlinear, time-varying relationships parameterized by learnable parameters $\mathbf{\Theta}_{ode}$. This framework unifies interpretable, physics-based principles with flexible, data-driven learning to robustly capture complex coupled interactions.*

To design a **Generalizable GraphODE** that effectively captures graph rationalization and facilitates generalization, we analyze its underlying mechanism using a **Structural Causal Model (SCM)** (Pearl et al., 2000; Pearl, 2016). The SCM framework enables us to systematically understand the causal relationships within coupled dynamical systems.

Below, we detail the key variables and their interactions (subscript $i$ omitted for brevity):

- $s \leftarrow x \rightarrow h(t_0)$: *Initial states* $x$ consist of two disjoint components: the **static variable** $s$, which encapsulates time-invariant properties (e.g., material properties, environmental conditions), and the **dynamic variable** $h(t_0)$, representing the system's initial dynamic state.

- $h(t) \xrightarrow{\partial h(t)/\partial t} h(t+1)$: The *state evolution process* describes how the dynamic variable $h(t)$ evolves over time, governed by a differential equation that captures temporal dependencies and interactions between entities.

From the SCM, two *backdoor paths* are identified that introduce spurious correlations, hindering generalization:

❶ $h(t_0) \leftarrow x \rightarrow s \rightarrow h(t) \rightarrow h(t+1)$ *Static states $s$ act as a confounder* between the initial state $h(t_0)$ and the evolving state $h(t)$. Static variables such as material properties should only influence initialization but not directly affect dynamic evolution. If improperly incorporated into the differential process, these variables can reduce generalization ability in scenarios with different static conditions.

❷ $h(t) \rightarrow \eta(t) \rightarrow h(t+1)$ : The *coupling factor* $\eta(t)$, derived from $h(t)$, acts as a confounder between the current state $h(t)$ and the next state $h(t+1)$. If $\eta(t)$ captures domain-specific contextual patterns (*e.g.*, particular coupling strengths in training data), it can lead to biased dynamics. Based on the discussion, we present the following principle for designing Generalizable GraphODE:

> **Generalizable GraphODE Design Principle**: **Disentanglement**: Effectively separate static variables from dynamic states, ensuring static properties influence only initialization without confounding dynamic evolution. **Regularization**: Address spurious correlations in the coupling relationship, ensuring context-invariant interactions aligned with universal physical rules. **Performance**: Achieve robust generalization both within the training distribution and out-of-distribution.

In the following sections, we will elaborate on how `GREAT` adheres to these principles, effectively severing spurious correlations through disentanglement and regularization.

## 3. Methodology

### 3.1. Framework Overview

The proposed `GREAT` framework achieves generalization in coupled dynamical systems through two core innovations: The Dynamic-Static Equilibrium Decoupler eliminates static-dynamic entanglement during state initialization via orthogonal subspace projections, decomposing system states into independent static attributes (*e.g.*, material properties) and dynamic patterns (*e.g.*, velocity fields). Comple-

menting this, the Causal Mediation for Coupled Dynamics addresses context-specific coupling biases through variational estimation of latent causal mediators, enabling the GraphODE to learn interaction dynamics governed by universal physical laws rather than observational artifacts. The illustration of the overall framework is detailed in Figure 3.

### 3.2. Dynamic-Static Equilibrium Decoupler

**Dynamic-Static Bilinear Orthogonal Projections.** Given the historical observation input $x_i(t_{-M} : t_{-1})$, we first encode it into a latent state $z_i = f_\theta(x_i) \in \mathbb{R}^{M \times d}$, where the encoder $f_\theta$ is implemented as a two-layer multilayer perceptron (MLP). The latent state $z_i$ captures both the temporal evolution (dynamic) and invariant contextual properties (static) of the input. To disentangle these components, we decompose $z_i$ into two orthogonal parts:

$$z_i = o_i + s_i, \quad o_i \perp s_i. \tag{5}$$

This decomposition separates $z_i$ into a **dynamic component** ($o_i$) and a **static component** ($s_i$), corresponding to the system's time-varying and invariant properties, respectively. To ensure trainability and adaptability, we introduce two learnable subspaces: the **static subspace** $\mathbf{S}_{\text{static}}$ and the **dynamic subspace** $\mathbf{S}_{\text{dynamic}}$, parameterized as:

$$\mathbf{S}_{\text{static}} = \text{EMBEDDING}_{\text{static}}(p, p),$$
$$\mathbf{S}_{\text{dynamic}} = \text{EMBEDDING}_{\text{dynamic}}(p, p), \tag{6}$$

where $\mathbf{S}_{\text{static}} = \{\mathbf{s}_{\text{static},1}, \mathbf{s}_{\text{static},2}, \ldots, \mathbf{s}_{\text{static},p}\}$ and $\mathbf{S}_{\text{dynamic}} = \{\mathbf{s}_{\text{dynamic},1}, \mathbf{s}_{\text{dynamic},2}, \ldots, \mathbf{s}_{\text{dynamic},p}\}$. Here, $p$ denotes the number of basis vectors in each subspace, with $\mathbf{s}_{\text{static},a} \in \mathbb{R}^{1 \times p}$ and $\mathbf{s}_{\text{dynamic},b} \in \mathbb{R}^{1 \times p}$ representing the $a$-th and $b$-th basis vectors of the static and dynamic subspaces, respectively. These subspaces are optimized during training to learn disentangled representations of $z_i$. We project $z_i$ onto $\mathbf{S}_{\text{static}}$ and $\mathbf{S}_{\text{dynamic}}$ to obtain $s_i$ and $o_i$:

$$s_i = \left\|_{a=1}^{p} \frac{\mathbf{s}_{\text{static},a} \cdot z_i^\top}{\|\mathbf{s}_{\text{static},a}\|_2} \cdot \frac{\mathbf{s}_{\text{static},a}}{\|\mathbf{s}_{\text{static},a}\|_2}\right.,$$
$$o_i = \left\|_{b=1}^{p} \frac{\mathbf{s}_{\text{dynamic},b} \cdot z_i^\top}{\|\mathbf{s}_{\text{dynamic},b}\|_2} \cdot \frac{\mathbf{s}_{\text{dynamic},b}}{\|\mathbf{s}_{\text{dynamic},b}\|_2}\right.. \tag{7}$$

Here, $s_i \in \mathbb{R}^{M \times d}$ and $o_i \in \mathbb{R}^{M \times d}$ denote the projections of $z_i$ onto the static and dynamic subspaces, respectively. The operator $\|$ denotes concatenation, and $\|\cdot\|_2$ is the Euclidean norm. While $s_i$ and $o_i$ are computed separately, orthogonality between the subspaces is not inherently guaranteed. To enforce this, we introduce an orthogonality loss $\mathcal{L}_o$:

$$\mathcal{L}_o = \sum_{a=1}^{p} \sum_{b=1}^{p} \text{abs}\left(\frac{\mathbf{s}_{\text{static},a}}{\|\mathbf{s}_{\text{static},a}\|_2} \cdot \left(\frac{\mathbf{s}_{\text{dynamic},b}}{\|\mathbf{s}_{\text{dynamic},b}\|_2}\right)^\top\right), \tag{8}$$

where $\text{abs}(\cdot)$ denotes the absolute value. Minimizing $\mathcal{L}_o$ reduces the cosine similarity between the subspaces, ensuring mutual orthogonality. This prevents interference between static and dynamic properties during prediction, enhancing both robustness and interpretability.

**Dynamic Hawkes Process Augmentation.** To further en-

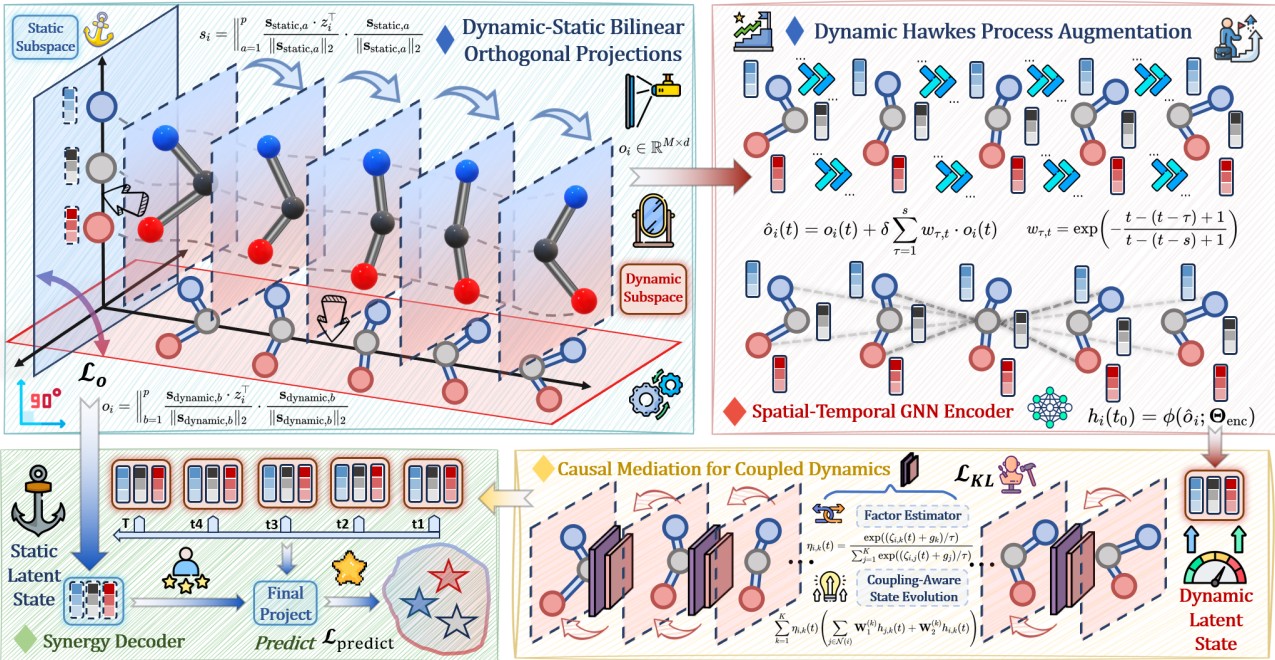

*Figure 3.* **Architecture illustration** of Generalizable GraphODE with disentanglement and regularization. Best viewed in color.

hance the disentangled dynamic representation $o_i$, we propose the Dynamic Hawkes Process Augmentation (DHPA), which leverages the self-exciting nature of the Hawkes process (Okawa et al.; Han et al.; Lin et al., 2021) to capture temporal dependencies in complex dynamical systems better. In the context of dynamic systems, states at a given time are rarely independent—they are influenced by prior states, often in a cascading manner. For example, forces propagate dynamically in physical systems, creating continuous interactions across time. While the disentangled representation $o_i$ captures immediate temporal patterns, it does not fully account for the compounding influence of past states. DHPA addresses this limitation by explicitly modeling these temporal interactions, enriching the dynamic representation and enabling it to capture both short- and long-term dependencies inherent in physical systems. Formally, the augmented representation at time $t$, $\hat{o}_i(t) \in \mathbb{R}^d$:

$$\hat{o}_i(t) = o_i(t) + \delta \sum_{\tau=1}^{s} w_{\tau,t} \cdot o_i(t), \quad (9)$$

where $s$ is the historical window size determining the temporal range of influence, and $\delta$ is a learnable parameter controlling the augmentation strength. The influence weight $w_{\tau,t}$, which quantifies the contribution of the past state $o_i(t-\tau)$ to the current state $o_i(t)$, is defined as:

$$w_{\tau,t} = \exp\left(-\frac{t - (t-\tau) + 1}{t - (t-s) + 1}\right), \quad (10)$$

ensuring an exponentially decaying influence from more distant past states, where more recent states exert stronger influence. By summing over this weighted contribution of past states, DHPA explicitly integrates the self-exciting and decaying temporal dynamics into $o_i$, augmenting the

representation with richer temporal dependencies:

$$\hat{o}_i = \{\hat{o}_i(-M), \hat{o}_i(-M+1), \dots, \hat{o}_i(-1)\}. \quad (11)$$

This augmentation offers several advantages: ❶ It explicitly captures cascading effects and long-term dependencies in dynamic observations, which are essential for understanding physical systems governed by differential equations. ❷ By integrating temporal interactions, it smooths noisy fluctuations and enhances robustness, ensuring that the representation remains stable across varying system conditions. ❸ The learned influence weights $w_{\tau,t}$ provide interpretable insights into the temporal structure of the system, revealing how past states influence future behavior.

**Dynamic-Static State Initialization.** To initialize the dynamic state for GraphODE and effectively learn spatial-temporal relationships among agents, we follow the methodology in (Huang et al., 2021; 2024b; Luo et al., 2024). We adopt a spatio-temporal GNN $\phi(\cdot)$, parameterized by $\Theta_{\text{enc}}$, to compress historical trajectories into latent representations: $h_i(t_0) = \phi(\hat{o}_i; \Theta_{\text{enc}})$. (For details about the spatio-temporal GNN $\phi(\cdot)$, see Appendix D.) For the static state, we directly apply a global pooling mechanism to summarize stationary characteristics, capturing unchanging information over time in $s_i$: $s_i = \frac{1}{M} \sum_{t=1}^{M} s_i(t)$. After obtaining the output $h_i(T)$ at time $T$ through dynamic evolution (as described in Sec. 3.3), we propose a synergy decoder to seamlessly combine these two components in the final output stage:

$$\hat{x}_i(T) = f_{\text{dec}}(h_i(T), s_i(T); \Theta_{\text{dec}}). \quad (12)$$

### 3.3. Causal Mediation for Coupled Dynamics

**Analysis on State Evolution.** After obtaining the separated dynamic state, we analyze how to design the generalized

GraphODE and sever spurious relationships ❷. The distribution process can be defined as $p(h_i(t+1)|h_i(t), \eta_i(t), \mathcal{G})$, governed by the differential process in Equation (4). Considering the impact of coupling factors, the generative form can be expressed as:

$$\mathbb{E}_{p(\eta_i(t)|h_i(t))} \left[ p(h_i(t+1)|h_i(t), \eta_i(t), \mathcal{G}) \right]. \quad (13)$$

While the generative distribution $p(h_i(t+1)|h_i(t), \eta_i(t), \mathcal{G})$ provides a principled framework for modeling state transitions, $\eta_i(t)$ can act as a confounding variable, introducing biases. Specifically, the distribution $p(\eta_i(t)|h_i(t))$ serves as a prior for coupling dynamics. However, relying on it biases the model toward the training data distribution, entangling the mapping from $h_i(t)$ to $h_i(t+1)$ with $\eta_i(t)$ and creating spurious correlations, as illustrated in Figure 2. These context-specific influences limit the model's ability to generalize to unseen scenarios, as coupled systems generally adhere to fundamental physical laws, such as the conservation of energy, which transcend specific coupling factors.

**Deconfounding Coupled Evolution.** To address the confounding influence of $\eta_i(t)$, we propose a causal regularization method. Inspired by deconfounded learning (Wu et al., 2024a), we aim to compute $\log p(y_i(T)|do(h_i(t_0)), \mathcal{G})$. The causal intervention $do(h_i(t_0))$ eliminates the spurious correlation between the coupling factor $\eta_i(t)$ and the state transition dynamics, enabling the model to focus on the invariant relationship between $h_i(t)$ and $h_i(t+1)$ that holds across domains. However, due to the unobservable nature of $\eta_i(t)$, we cannot directly compute this quantity. Instead, we leverage a variational distribution $q(\eta_i(t)|h_i(t))$, allowing the model to approximate the interventional likelihood $\log p(y_i(T)|do(h_i(t_0)), \mathcal{G})$ in a tractable manner:

**Theorem 3.1.** *Consider the state evolution process $p(h_i(t+1)|h_i(t), \eta_i(t), \mathcal{G})$ with the coupling factor $\eta_i(t)$ as a latent confounder. The regularization likelihood $\log p(y_i(T)|do(h_i(t_0)), \mathcal{G})$ can be bounded as:*

$$\log p(y_i(T)|do(h_i(t_0)), \mathcal{G}) \geq \log p(y_i(T)|h_i(T), s_i; \Theta_{dec})p(s_i) +$$

$$\sum_{t=0}^{T} \mathbb{E}_{q(\eta_i(t)|h_i(t))} \left[ \log \sum_{h_i(t+1)} p(h_i(t+1)|h_i(t), \eta_i(t), \mathcal{G}; \Theta_{ode}) \right]$$

$$- \sum_{t=0}^{T} KL \left( q(\eta_i(t)|h_i(t)) \parallel p(\eta_i(t)) \right), \quad (14)$$

*where $p(\eta_i(t))$ is a context-agnostic prior over the coupling factor, and $q(\eta_i(t)|h_i(t))$ is the variational approximation of the posterior for $\eta_i(t)$.*

The proof of this bound is provided in Appendix B. Here, $p(\eta_i(t))$ is typically assumed to be a uniform prior, $p(\eta_i(t)) = 1/K$, to ensure that $\eta_i(t)$ remains agnostic to domain-specific contexts. The causal mediation objective introduces a scaling term $\frac{p(\eta_i(t))}{q(\eta_i(t)|h_i(t))}$, which adjusts the influence of the coupling factor $\eta_i(t)$ on the learning process.

By incorporating a context-independent prior $p(\eta_i(t))$, the scaling term prevents the model from overfitting to context-specific correlations introduced by $q(\eta_i(t)|h_i(t))$, enhancing its ability to handle distributional shifts. Furthermore, the scaling term reduces the influence of frequently occurring coupling patterns and increases the importance of rarer ones, promoting the model's focus on invariant dynamics that hold across different environments. By approximating $p(y_i(t)|do(h_i(t_0)), \mathcal{G})$ through the scaling term, the model aligns with invariant principles governing coupled dynamics, such as conservation laws. This enables the ODE solver to learn stable dynamics from $h_i(t)$ to $h_i(t+1)$, facilitating robust predictions and ensuring unbiased dynamics.

**Factor Estimator.** To tackle the confounding effect introduced by the coupling factor $\eta_i(t)$, we introduce a mechanism that explicitly estimates $\eta_i(t)$ based on the disentangled dynamic state $h_i(t)$. We define $\eta_i(t) \in \mathbb{R}^K$ as a $K$-dimensional representation of the inferred coupling factor for node $i$ at time $t$. Instead of directly assigning $\eta_i(t)$, it is modeled as a probabilistic variable conditioned on the node's dynamic state $h_i(t)$. Formally, we parameterize the distribution of $\eta_i(t)$ as $\zeta_i(t) = \text{Softmax}(\mathbf{W}_\eta h_i(t))$, where $\zeta_i(t) \in \mathbb{R}^K$ represents the probability distribution over $K$ possible coupling patterns, $\mathbf{W}_\eta \in \mathbb{R}^{d \times K}$ is a learnable parameter matrix, and $d$ is the dimension of $h_i(t)$. The coupling factor $\eta_i(t)$ is then sampled using the Gumbel-Softmax approach (Jang et al., 2016; Shah et al., 2024) to ensure differentiability during backpropagation:

$$\eta_{i,k}(t) = \frac{\exp\left((\zeta_{i,k}(t) + g_k)/\tau\right)}{\sum_{j=1}^{K} \exp\left((\zeta_{i,j}(t) + g_j)/\tau\right)}, \quad g_k \sim \text{Gumbel}(0,1), \quad (15)$$

where $g_k$ is noise sampled from the Gumbel distribution, and $\tau$ is the temperature parameter controlling the smoothness.

**Coupling-Aware State Evolution in GraphODE.** To better model complex interactions in dynamical systems, we decompose the dynamic state $h_i(t)$ into sub-states $h_{i,k}(t)$, each representing a specific coupling pathway $\eta_{i,k}(t)$. This allows the model to capture distinct periodic and oscillatory behaviors, such as synchronization and resonance. The state is represented using a trigonometric basis, where $h_{i,k}(t)$ is expressed as sine and cosine transformations:

$$h_{i,2m-1}(t) = \sin(m \cdot h_i(t)), \quad h_{i,2m}(t) = \cos(m \cdot h_i(t)). \quad (16)$$

Each sub-state $h_{i,k}(t)$ interacts with its respective coupling factor $\eta_{i,k}(t)$, allowing independent modulation of state evolution for each mode. The inferred coupling factor $\eta_i(t)$ influences state transitions, dynamically adjusting sub-state evolution. Unlike traditional static coupling rules, our method adapts each sub-state based on its associated coupling factor. The overall state evolution is modeled as:

$$\frac{\partial h_i(t)}{\partial t} = \sum_{k=1}^{K} \eta_{i,k}(t) \left( \sum_{j \in \mathcal{N}(i)} \mathbf{W}_1^{(k)} h_{j,k}(t) + \mathbf{W}_2^{(k)} h_{i,k}(t) \right), \quad (17)$$

| Methods | SPRING | | | | CHARGED | | | | PENDULUM | | | |
| --- | --- | --- | --- | --- | --- | --- | --- | --- | --- | --- | --- | --- |
| | In-Distribution | | Out-of-Distribution | | In-Distribution | | Out-of-Distribution | | In-Distribution | | Out-of-Distribution | |
| | RMSE | MAPE | RMSE | MAPE | RMSE | MAPE | RMSE | MAPE | RMSE | MAPE | RMSE | MAPE |
| VAR | $9.881_{\uparrow3.210}$ | $24.55_{\uparrow18.01}$ | $13.11_{\uparrow5.729}$ | $25.67_{\uparrow18.13}$ | $11.32_{\uparrow3.883}$ | $30.20_{\uparrow13.45}$ | $18.45_{\uparrow8.220}$ | $27.88_{\uparrow4.090}$ | $36.72_{\uparrow19.26}$ | $42.31_{\uparrow3.350}$ | $42.77_{\uparrow8.550}$ | $65.02_{\uparrow10.10}$ |
| LSTM | $7.992_{\uparrow1.321}$ | $20.31_{\uparrow4.232}$ | $10.22_{\uparrow2.839}$ | $15.09_{\uparrow2.447}$ | $7.432_{\uparrow0.012}$ | $22.63_{\uparrow5.121}$ | $24.66_{\uparrow14.43}$ | $31.45_{\uparrow7.680}$ | $30.64_{\uparrow13.18}$ | $40.22_{\uparrow1.260}$ | $37.08_{\uparrow2.860}$ | $60.74_{\uparrow5.820}$ |
| LatentODE | 6.671 | 6.544 | 7.381 | 7.542 | 7.444 | 16.75 | 10.23 | 23.77 | 17.46 | 38.96 | 34.22 | 54.92 |
| HODEN | $5.029_{\downarrow1.642}$ | $8.327_{\uparrow1.783}$ | $8.139_{\uparrow0.758}$ | $10.09_{\uparrow2.451}$ | $5.892_{\downarrow1.552}$ | $11.23_{\downarrow5.520}$ | $5.107_{\downarrow5.123}$ | $13.67_{\downarrow10.10}$ | $27.68_{\uparrow10.22}$ | $40.55_{\uparrow1.590}$ | $39.45_{\uparrow5.230}$ | $60.22_{\uparrow5.300}$ |
| TRS-ODEN | $5.456_{\downarrow1.215}$ | $4.801_{\downarrow1.743}$ | $7.120_{\downarrow0.261}$ | $8.620_{\uparrow1.078}$ | $4.601_{\downarrow2.843}$ | $10.45_{\downarrow6.303}$ | $6.452_{\downarrow3.781}$ | $18.22_{\downarrow5.550}$ | $38.92_{\uparrow21.46}$ | $46.89_{\uparrow7.930}$ | $38.74_{\uparrow4.520}$ | $67.33_{\uparrow12.41}$ |
| LG-ODE | $4.307_{\downarrow2.364}$ | $2.918_{\downarrow3.626}$ | $4.346_{\downarrow3.035}$ | $6.112_{\downarrow1.430}$ | $2.883_{\downarrow4.561}$ | $6.430_{\downarrow10.32}$ | $3.145_{\downarrow7.085}$ | $10.09_{\downarrow13.69}$ | $11.43_{\downarrow6.030}$ | $23.90_{\downarrow15.06}$ | $19.35_{\downarrow14.87}$ | $33.70_{\downarrow21.22}$ |
| SocialODE | $4.452_{\downarrow2.219}$ | $3.520_{\downarrow3.024}$ | $4.902_{\downarrow2.479}$ | $5.221_{\downarrow2.321}$ | $2.756_{\downarrow4.688}$ | $6.788_{\downarrow9.962}$ | $3.001_{\downarrow7.232}$ | $8.452_{\downarrow15.32}$ | $13.72_{\downarrow3.740}$ | $26.33_{\downarrow12.63}$ | $24.04_{\downarrow10.18}$ | $37.72_{\downarrow17.02}$ |
| PGODE | $3.901_{\downarrow2.770}$ | $2.562_{\downarrow3.982}$ | $4.021_{\downarrow3.360}$ | $5.772_{\downarrow1.770}$ | $2.564_{\downarrow4.880}$ | $6.503_{\downarrow10.24}$ | $2.904_{\downarrow7.326}$ | $9.192_{\downarrow14.58}$ | $10.92_{\downarrow6.540}$ | **$22.71_{\downarrow16.25}$** | $20.19_{\downarrow14.03}$ | $32.90_{\downarrow22.02}$ |
| TREAT | $3.892_{\downarrow2.779}$ | $2.748_{\downarrow3.796}$ | $4.213_{\downarrow3.168}$ | $5.234_{\downarrow2.308}$ | $2.910_{\downarrow4.534}$ | $7.234_{\downarrow9.516}$ | $3.236_{\downarrow7.000}$ | $10.42_{\downarrow13.35}$ | $11.96_{\downarrow5.500}$ | $28.91_{\downarrow14.05}$ | $22.34_{\downarrow12.08}$ | $34.21_{\downarrow20.71}$ |
| GREAT | **$3.687_{\downarrow2.984}$** | **$2.348_{\downarrow4.196}$** | **$3.619_{\downarrow3.762}$** | **$4.521_{\downarrow3.021}$** | **$2.337_{\downarrow5.107}$** | **$6.042_{\downarrow10.71}$** | **$2.690_{\downarrow7.540}$** | **$8.322_{\downarrow15.45}$** | **$10.34_{\downarrow7.120}$** | $23.44_{\downarrow15.52}$ | **$18.24_{\downarrow16.05}$** | **$30.44_{\downarrow24.48}$** |

*Table 1.* **Comparison with the counterparts** on three datasets with RMSE and MAPE ($10^{-1}$). Best in bold and second with underline.

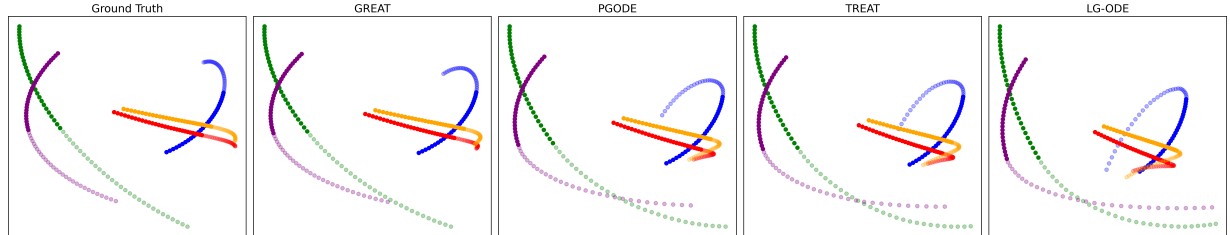

*Figure 4.* **Visualization of predicted trajectories** (Semi-transparent) given observation trajectories (Solid) in the *SPRING* OOD scenario.

where $\mathcal{N}(i)$ denotes node $i$'s neighbors, and $\mathbf{W}_1^{(k)}, \mathbf{W}_2^{(k)}$ are trainable weight matrices for the $k$-th coupling pathway. This trigonometric decomposition ensures that GraphODE can effectively capture periodic and interaction-specific dynamics in a flexible and interpretable way.

### 3.4. Training Objective

The final training objective integrates prediction accuracy, disentanglement, and regularization into a unified loss:

$$\mathcal{L} = \mathcal{L}_{\text{predict}} - \mathcal{L}_{\text{KL}} + \lambda_o \mathcal{L}_o, \quad (18)$$

where $\mathcal{L}_{\text{predict}}$ ensures accurate predictions, $\mathcal{L}_o$ enforces orthogonality between static and dynamic subspaces, and $\mathcal{L}_{\text{KL}}$ regularizes the coupling factor $\eta_i(t)$ estimation. $\mathcal{L}_{\text{predict}}$ and $\mathcal{L}_{\text{KL}}$ together form the core of the evidence lower bound (ELBO), where $\mathcal{L}_{\text{predict}}$ corresponds to the reconstruction likelihood, and $\mathcal{L}_{\text{KL}}$ imposes a regularization. Specifically, $\mathcal{L}_{\text{KL}}$, derived from Equation (14), can be calculated as:

$$\mathcal{L}_{\text{KL}} = \frac{1}{N} \sum_{i=1}^{N} \sum_{t=0}^{T} \text{KL}\big(q(\eta_i(t) \mid h_i(t)) \parallel p(\eta_i(t))\big), \quad (19)$$

this ensures that $\eta_i(t)$ is disentangled from spurious correlations and aligns with context-invariant principles.

## 4. Experiment

We comprehensively evaluate **GREAT** through four axes: **Q1** (Superiority), **Q2** (Resilience), **Q3** (Effectiveness), and **Q4** (Sensitivity). **Q1-Q3** are illustrated in Sec. 4.2-Sec. 4.4, and sensitivity analyses (**Q4**) can be found in the Appendix F.

### 4.1. Experimental Setup

**Coupled Dynamical Systems Datasets.** We evaluate **GREAT** using three coupled dynamical systems datasets: *SPRING*, *CHARGED*, and *PENDULUM*, which model the dynamics of physical systems with complex interdependencies. For evaluation, we adopt two metrics: RMSE (Root Mean Square Error) and MAPE (Mean Absolute Percentage Error). We assess **GREAT** under both in-distribution (ID) and out-of-distribution (OOD) settings, where the OOD setting modifies the test dataset's initial conditions (*e.g.*, velocity, position) to test the model's ability to generalize to unseen scenarios. Further details are provided in Appendix A. The code is available at https://github.com/GuanchengWan/GREAT.

**Counterparts.** We compare ours against traditional approaches and several SOTA Neural ODE methods including VAR (Song et al., 2020), LSTM (Sesti et al., 2021), LatentODE (Rubanova et al., 2019), HODEN (Greydanus et al., 2019), TRS-ODEN (Huh et al., 2020), LG-ODE (Huang et al., 2020), SocialODE (Wen et al., 2022), PGODE (Luo et al., 2024), TREAT (Huang et al., 2024b).

### 4.2. Superiority (Q1)

To assess the performance of **GREAT**, we evaluate it in both ID and OOD settings, as shown in Tab. 1. Several observations can be made (**Obs.**): **Obs. ❶** Traditional methods, such as LSTM and VAR, often perform worse than Neural ODE-based deep learning methods, highlighting the advan-

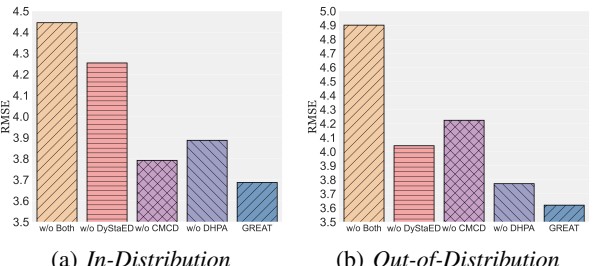

(a) *In-Distribution*            (b) *Out-of-Distribution*

*Figure 5*. **Analysis on proposed components** for both ID and OOD settings on *SPRING*. Please see details in Sec. 4.4.

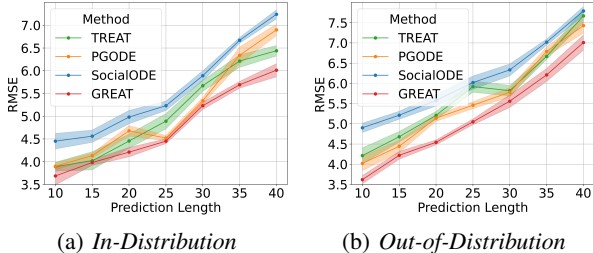

(a) *In-Distribution*            (b) *Out-of-Distribution*

*Figure 6*. **Analysis on prediction length** for both ID and OOD settings on *SPRING*. Please see details in Sec. 4.3.

tages of incorporating dynamic system modeling for complex tasks. **Obs. ❷** `GREAT` consistently achieves the best performance across both RMSE and MAPE metrics, outperforming all baselines in both ID and OOD settings. This demonstrates its robustness and adaptability. Specifically, `GREAT` achieves an RMSE of 3.687 and a MAPE of 2.348 on the *SPRING* dataset in the ID setting, outperforming all other methods, including traditional approaches, as well as state-of-the-art Neural ODE models such as LatentODE and TREAT. **Obs. ❸** In OOD scenarios, `GREAT` exhibits even greater performance improvements compared to the ID setting, demonstrating that the correct decoupling and regularization techniques help stabilize the model and enhance its generalization ability.

Furthermore, we visualized the model's predicted trajectories (semi-transparent) alongside the given observation trajectories (solid) in the *SPRING* OOD scenario. As shown in Figure 4, `GREAT` more accurately fits the ground truth compared to other methods. The solid ground truth trajectory exhibits smooth, consistent behavior, with `GREAT` closely following the true dynamics.

### 4.3. Resilience (Q2)

As shown in Figure 6, `GREAT` consistently achieves the best performance across various prediction lengths, demonstrating its resilience as the prediction horizon extends. In both ID and OOD scenarios, `GREAT` maintains a lower RMSE compared to other methods, underscoring its robustness in handling long-term predictions. Particularly in the OOD setting (as shown in Figure 6 (b)), `GREAT`'s performance remains stable and competitive, outperforming methods such as TREAT and PGODE. This indicates that `GREAT` can ef-

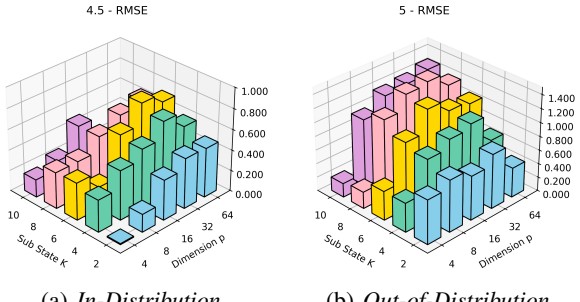

(a) *In-Distribution*            (b) *Out-of-Distribution*

*Figure 7*. **Analysis on Substate** $K$ **and dimension** $p$ for both ID and OOD settings on *SPRING*. Please see details in Appendix F.

fectively generalize and sustain performance, even under challenging conditions with varying prediction lengths.

### 4.4. Effectiveness (Q3)

We evaluate the effectiveness of two key components in `GREAT`: DyStaED and CMCD. As shown in Figure 5, both components significantly improve model performance. `GREAT` enhances the disentangling of dynamic and static components, addressing spurious dependencies and enabling better generalization under new conditions. DHPA further improves this disentanglement, encouraging the characteristics of dynamic representation and separation. Meanwhile, CMCD contributes to modeling coupling dynamics, allowing the model to capture system's robust interdependencies. When both DyStaED and CMCD are combined, performance reaches its peak, as evidenced by the lowest RMSE across both ID and OOD settings. Notably, CMCD's impact is more pronounced in OOD scenarios (Figure 5 (b)). This indicates that learning universal coupled dynamics is crucial for improving generalization to unseen scenarios.

## 5. Conclusion

In this paper, we propose a novel framework, `GREAT`, to enhance the generalization capabilities of GraphODE models for coupled dynamical systems. Leveraging insights from Structural Causal Models, we identify two key challenges: the entanglement of static attributes with dynamic states during initialization and the reliance on context-specific coupling patterns that hinder performance in unseen scenarios. To address these issues, we introduce the *Dynamic-Static Equilibrium Decoupler (DyStaED)*, which disentangles static and dynamic states through orthogonal subspace projections, ensuring robust initialization. Additionally, we propose the *Causal Mediation for Coupled Dynamics (CMCD)*, which uses variational inference to estimate latent causal factors, mitigating spurious correlations and promoting universal coupling dynamics. Extensive experiments on diverse coupled dynamical systems demonstrate that `GREAT` outperforms state-of-the-art methods in both in-distribution and out-of-distribution settings. Our `GREAT` provides a ro-

bust solution for modeling complex systems and highlights the importance of disentanglement and regularization for generalization across dynamic environments.

## Acknowledgement

This work was partially supported by NSF 2200274, NSF 2106859, NSF 2312501, NSF 2211557, NSF 2119643, NSF 2303037, DARPA HR00112490370, NIH U54HG012517, NIH U24DK097771, NIH U54OD036472, SRC JUMP 2.0 Center, Amazon Research Awards, Snapchat Gifts, NEC, and Optum.

## Impact Statement

This paper presents work whose goal is to advance the field of Machine Learning. There are many potential societal consequences of our work, none of which we feel must be specifically highlighted here.

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

# A. Datasets and Settings

The details of these datasets are described below: The ***SPRING*** dataset simulates a system of interconnected springs governed by Hooke's law. Each sample contains 5 interacting springs with properties such as elasticity coefficients, initial positions, velocity, and acceleration. The system parameters include box size ($\alpha$), initial velocity norm ($\beta$), interaction strength ($\gamma$), and spring connection probability ($\delta$). The ***CHARGED*** dataset simulates electromagnetic phenomena, where particles interact through Coulomb forces. Similar to the ***SPRING*** dataset, each sample contains 5 particles with properties such as charge, mass, and initial velocity. The system parameters include box size ($\alpha$), initial velocity norm ($\beta$), interaction strength ($\gamma$), and charge probability ($\delta$). Notably, particles in the ***CHARGED*** system attract or repel with equal probability, while the ***SPRING*** system has unequal probabilities for spring connections. The ***PENDULUM*** dataset simulates a system of connected pendulums, where each pendulum is governed by the laws of classical mechanics. Each sample contains 3 connected pendulums with properties such as stick length, mass, and initial angular positions. The system parameters include: ,$L$: Length of the pendulum sticks (in meters), $m$: Mass of the pendulum sticks (in kilograms), $\sigma_{\text{loc}}$: Standard deviation of initial angular positions (in radians), $v_{\text{norm}}$: Norm of initial angular velocities (in radians per second).

The datasets are split into training, validation, and test sets, with additional out-of-distribution (OOD) test sets to evaluate model generalization. The parameter ranges for ID (in-distribution) and OOD (out-of-distribution) settings are as follows: The datasets are generated using a physics-based simulation framework, where the system parameters are sampled from

| Parameter | *SPRING* | *CHARGED* | *PENDULUM* |
|---|---|---|---|
| $\alpha$ (box size) | [4.9, 5.1] (ID) | [4.9, 5.1] (ID) | - |
| $\beta$ (initial velocity norm) | [0.49, 0.51] (ID) | [0.49, 0.51] (ID) | - |
| $\gamma$ (interaction strength) | [0.09, 0.11] (ID) | [0.9, 1.1] (ID) | - |
| $\delta$ (spring/charge probability) | [0.49, 0.51] (ID) | [0.49, 0.51] (ID) | - |
| $L$ (stick length) | - | - | [0.9, 1.1] (ID) |
| $m$ (stick mass) | - | - | [0.9, 1.1] (ID) |
| $\sigma_{\text{loc}}$ (position std) | - | - | [0.09, 0.11] (ID) |
| $v_{\text{norm}}$ (velocity norm) | - | - | [0.49, 0.51] (ID) |
| **OOD Test Set** | $\alpha \in [4.5, 5.5]$ 
 $\beta \in [0.45, 0.55]$ 
 $\gamma \in [0.05, 0.15]$ 
 $\delta \in [0.45, 0.55]$ | $\alpha \in [4.5, 5.5]$ 
 $\beta \in [0.45, 0.55]$ 
 $\gamma \in [0.5, 1.5]$ 
 $\delta \in [0.45, 0.55]$ | $L \in [0.5, 1.5]$ 
 $m \in [0.5, 1.5]$ 
 $\sigma_{\text{loc}} \in [0.05, 0.15]$ 
 $v_{\text{norm}} \in [0.45, 0.55]$ |
| **Number of Samples** | | | |
| Training Set | 5000 | 5000 | 5000 |
| Validation Set | 1000 | 1000 | 1000 |
| Test Set | 1000 | 1000 | 1000 |
| OOD Test Set | 1000 | 1000 | 1000 |

*Table 2.* Datasets and distributions of system parameters. For the OOD test set, there is at least one of the system parameters outside the range utilized for training.

the specified ranges. For the OOD test sets, at least one parameter is sampled outside the ID range to evaluate the model's generalization capability. The training, validation, and test sets are split as follows: 5000 samples for training, 1000 samples for validation, 1000 samples for testing, and 1000 samples for OOD testing. This setup ensures a rigorous evaluation of the model's performance in both in-distribution and out-of-distribution scenarios.

# B. Proof of Equation (14)

We first introduce the fundamental rules of do-calculus, which are crucial for understanding the causal relationships between variables in a directed acyclic graph (DAG). These rules allow us to manipulate and simplify expressions involving interventions and observations in causal models. We will then apply these rules to derive the causal intervention for causal inference in the context of a Structural Causal Model (SCM), as shown in Figure 2.

## B.1. Rules of Do-Calculus

Do-calculus is a formal system introduced by Pearl that provides a set of rules to handle interventional distributions. Let $G$ be a directed acyclic graph (DAG) representing the causal relationships between variables, and let $G_{do(x)}$ be the graph obtained by intervening on variable $X$. The graph $G_{do(x)}$ is identical to $G$ except for the removal of all arrows leading to $X$ from its parents. The nullified graph $G_{null(x)}$ is defined as the graph where all arrows from $X$ have been removed. We can now introduce three primary rules of do-calculus (Pearl et al., 2000; Pearl, 2016):

1. **Insertion/Deletion of Observations (Rule 1)**: This rule states that if $Y$ and $Z$ are independent given $X$ in the interventional graph $G_{do(x)}$, then we can remove $Z$ from the conditioning set:

$$P(y \mid do(x), z) = P(y \mid do(x)) \quad \text{if} \quad y \perp z \mid x \text{ in } G_{do(x)}.$$

2. **Action/Observation Exchange (Rule 2)**: This rule allows for exchanging an intervention on a variable $Z$ with an observation, provided that $Z$ is independent of other variables when conditioned on $X$ in the interventional graph $G_{do(x)}$ and the nullified graph $G_{null(z)}$:

$$P(y \mid do(x), do(z)) = P(y \mid do(x), z) \quad \text{if} \quad y \perp z \mid x \text{ in } G_{do(x)}, G_{null(z)}.$$

3. **Insertion/Deletion of Actions (Rule 3)**: This rule asserts that if $Y$ and $Z$ are independent given $X$ in the interventional graph $G_{do(x)}$ and the graph where both $X$ and $Z$ are intervened, we can remove the intervention on $Z$ from the conditioning set:
$$P(y \mid do(x), do(z)) = P(y \mid do(x)) \quad \text{if} \quad y \perp z \mid x \text{ in } G_{do(x)}, G_{do(z)}.$$

These rules allow us to manipulate interventional distributions and simplify expressions involving interventions in causal graphs. They are particularly useful for deriving causal effects from observational data when randomization or controlled experiments are not feasible.

## B.2. Interventional Likelihood Derivation

We derive the interventional likelihood for the coupled dynamical system by combining causal calculus with neural ODEs. Let $y_i(T)$ denote the observational target at terminal time $T$, $h_i(t)$ the latent states governed by physical interaction dynamics, and $\eta_i(t)$ the latent coupling factors mediating state transitions. The derivation establishes a deconfounded learning objective through the following key transformations:

$$
\begin{aligned}
&\log p(y_i(T)|do(h_i(t_0)), \mathcal{G}) \\
&= \log \left[ p(y_i(T)|h_i(T), s_i, do(h_i(t_0)); \Theta_{\text{dec}}) p(s_i) p(h_i(T)|do(h_i(t_0)), \mathcal{G}; \Theta_{\text{ode}}) \right] \quad (20)
\end{aligned}
$$

$$
= \log \left[ p(y_i(T)|h_i(T), s_i; \Theta_{\text{dec}}) p(s_i) \sum_{\eta_i} p(h_i(T)|do(h_i(t_0)), \eta_i, \mathcal{G}; \Theta_{\text{ode}}) p(\eta_i|do(h_i(t_0))) \right] \quad (21)
$$

$$
= \log \left[ p(y_i(T)|h_i(T), s_i; \Theta_{\text{dec}}) p(s_i) \sum_{\eta_i} p(h_i(T)|h_i(t_0), \eta_i, \mathcal{G}; \Theta_{\text{ode}}) p(\eta_i|do(h_i(t_0))) \right] \quad (22)
$$

$$
= \log \left[ p(y_i(T)|h_i(T), s_i; \Theta_{\text{dec}}) p(s_i) \sum_{\eta_i} p(h_i(T)|h_i(t_0), \eta_i, \mathcal{G}; \Theta_{\text{ode}}) p(\eta_i) \right] \quad (23)
$$

The derivation progresses through three crucial causal manipulations. First, we marginalize over the latent coupling factors $\eta_i$ while preserving the $do(h_i(t_0))$ intervention in both ODE dynamics and coupling factor distributions. Subsequently, we apply do-calculus Rule 2 to eliminate the intervention from $p(h_i(T)|do(h_i(t_0)), \eta_i, \mathcal{G})$ by leveraging the d-separation property in $G_{do(h_i(t_0))}$: given the parent states $\{h_i(t_0), \eta_i\}$, the intervention becomes conditionally independent of subsequent state evolution. Finally, Rule 3 justifies replacing $p(\eta_i|do(h_i(t_0)))$ with $p(\eta_i)$ since the coupling factors $\eta_i$ remain invariant to interventions on system states under our structural causal model.

To address the temporal confounding effects in coupled dynamics, we decompose the joint coupling factor distribution through Markovian factorization:

$$p(\eta_i(0), \ldots, \eta_i(T-1)) = \prod_{t=0}^{T-1} p(\eta_i(t))$$

Combined with the state evolution process $p(h_i(t+1)|h_i(t), \eta_i(t), \mathcal{G})$, we expand Equation (23) into path-space integrals over all possible latent trajectories:

$$
\begin{aligned}
& \log p(y_i(T)|h_i(T), s_i; \mathbf{\Theta}_{\text{dec}})p(s_i) \\
& \sum_{\substack{h_i(t_1),\ldots,h_i(T) \\ \eta_i(0),\ldots,\eta_i(T-1)}} \prod_{t=0}^{T-1} p(h_i(t+1)|h_i(t), \eta_i(t), \mathcal{G}; \mathbf{\Theta}_{\text{ode}})p(\eta_i(t)) \qquad (24) \\
& = \log p(y_i(T)|h_i(T), s_i; \mathbf{\Theta}_{\text{dec}})p(s_i) \\
& + \sum_{t=0}^{T-1} \log \sum_{h_i(t+1)} \sum_{\eta_i(t)} p(h_i(t+1)|h_i(t), \eta_i(t), \mathcal{G}; \mathbf{\Theta}_{\text{ode}})p(\eta_i(t)) \qquad (25)
\end{aligned}
$$

As shown in Equation (24), the temporal factorization enables decomposing the joint likelihood into products of Markov transitions governed by the coupled dynamics. The logarithmic transformation in Equation (25) further separates the objective into additive components across time steps, revealing the cumulative impact of coupling factors on system evolution.

To resolve the intractability induced by latent coupling factors, we introduce a variational posterior $q(\eta_i(t)|h_i(t))$ that approximates the true confounding distribution. Through importance sampling and Jensen's inequality, we derive the evidence lower bound:

$$
\begin{aligned}
& \log p(y_i(T)|h_i(T), s_i; \mathbf{\Theta}_{\text{dec}})p(s_i) \\
& + \sum_{t=0}^{T-1} \log \sum_{h_i(t+1)} \sum_{\eta_i(t)} p(h_i(t+1)|h_i(t), \eta_i(t), \mathcal{G}; \mathbf{\Theta}_{\text{ode}})\frac{p(\eta_i(t))p(\eta_i(t)|h_i(t))}{q(\eta_i(t)|h_i(t))} \qquad (26) \\
& \geq \log p(y_i(T)|h_i(T), s_i; \mathbf{\Theta}_{\text{dec}})p(s_i) \\
& + \sum_{t=0}^{T-1} \mathbb{E}_{q(\eta_i(t)|h_i(t))} \left[ \log \sum_{h_i(t+1)} p(h_i(t+1)|h_i(t), \eta_i(t), \mathcal{G}; \mathbf{\Theta}_{\text{ode}}) \right] \\
& - \sum_{t=0}^{T-1} \text{KL}\left( q(\eta_i(t)|h_i(t)) \,\|\, p(\eta_i(t)) \right) \qquad (27)
\end{aligned}
$$

The variational formulation in Equation (27) achieves three critical objectives: 1) The reconstruction term preserves observational consistency through the decoder $\mathbf{\Theta}_{\text{dec}}$; 2) The expectation term enforces physical plausibility of state transitions under the neural ODE constraints; 3) The KL divergence regularizes the coupling factor posterior against a domain-agnostic prior $p(\eta_i(t))$, effectively mitigating spurious correlations induced by context-specific coupling patterns. This regularization forces the model to disentangle invariant physical laws from transient coupling effects, as visualized in Figure 2.

By bridging causal intervention theory with variational inference, the derived bound enables robust learning of coupled dynamics that generalizes beyond observed coupling configurations. The complete implementation framework building upon this theoretical foundation is detailed in Sec. 3.3.

## C. Related Work

### C.1. Graph Neural Networks

Graph Neural Networks (GNNs) (Hamilton et al., 2017; Veličković et al., 2017; Kipf & Welling, 2017; Fu et al., 2022; Chen et al., 2024) are widely recognized for processing non-Euclidean data structures, including social network analysis and recommender systems (Xu et al., 2021; Cai et al., 2023; Liu et al., 2023; Chen et al., 2023). They update node representations by aggregating information from neighbors via message-passing (He et al., 2022a; You et al., 2020; Liu et al., 2022; Zhu et al., 2021; He et al., 2022b; Wan et al., 2025b). This approach enables GNNs to capture the complex dependencies (Zhang et al., 2024b;a) and structures inherent in graph data (Xia et al., 2021; Wan et al., 2024b; Zhang et al., 2024c; Huang et al., 2023a; Wan et al., 2024a). However, due to their unique message-passing mechanism, GNNs often learn patterns specific to the context of neighboring nodes. In coupled dynamic systems, this can lead to models that overfit to certain biases present in the training data, resulting in a loss of generalization ability under new conditions.

### C.2. Neural Ordinary Differential Equation

Neural Ordinary Differential Equation (ODE) is a class of deep learning models that treat the depth of a neural network as a continuous variable, enabling the modeling of complex, continuous-time dynamics. Introduced in (Chen et al., 2018), Neural ODEs parameterize the derivative of the hidden state using a neural network, allowing for the modeling of both static and dynamic systems (Chapfuwa et al.; Auzina et al.; Mei et al.). Building upon this framework, Graph Neural Ordinary Differential Equation (GraphODE) extend the concept to graph-structured data (Huang et al., 2020; 2021; Luo et al., 2023; Huang et al., 2023b; Qin et al., 2024; Huang et al., 2024a; Wan et al., 2025a). In (Poli et al., 2019), researchers formalized GraphODE as the continuous counterpart to GNN, where the input-output relationship is determined by a continuum of GNN layers. This approach blends discrete topological structures with differential equations, offering computational advantages and improved performance by leveraging the geometry of underlying dynamics (Wen et al., 2022; Huang et al., 2024b).

## D. Spatio-temporal GNN

Following the spatio-temporal GNN backbone (Huang et al., 2021; 2024b; Luo et al., 2024), the process begins with the construction of a temporal graph $G^{tem}$, where nodes represent individual observations, and edges encode both temporal and spatial relationships. Temporal edges connect consecutive observations of the same entity, while spatial edges link observations of different entities at the same timestamp. The adjacency matrix $\mathbf{A}^{tem}$ is defined as:

$$\mathbf{A}^{tem}(i^t, j^{t'}) = \begin{cases} w_{ij}^t & t = t', \\ 1 & i = j,\, t' = t+1, \\ 0 & \text{otherwise}, \end{cases} \tag{28}$$

where $i^t$ denotes the observation of entity $i$ at time $t$, and $w_{ij}^t$ represents weights derived from the spatial graph $G^t$. To capture temporal and spatial dependencies, information is propagated through the temporal graph using a message-passing mechanism. The representation of each node at layer $l$ is denoted as $\hat{o}_i^{(l)}(t)$, which is iteratively updated. Temporal patterns are incorporated by adding a temporal encoding $TE(t)$ to each node, where

$$TE(t)[2k] = \sin\left(\frac{t}{10000^{2k/d}}\right), \quad TE(t)[2k+1] = \cos\left(\frac{t}{10000^{2k/d}}\right),$$

with $d$ representing the feature dimension. The interaction between nodes is modeled using attention, computed as:

$$\alpha^{(l)}(i^t, j^{t'}) = \frac{\mathbf{A}^{tem}(i^t, j^{t'})}{\sqrt{d}} \left(\mathbf{W}_q \tilde{o}_i^{(l)}(t)\right)^T \left(\mathbf{W}_k \tilde{o}_j^{(l)}(t')\right), \tag{29}$$

where $\tilde{o}_i^{(l)}(t) = \hat{o}_i^{(l)}(t) + TE(t)$, and $\mathbf{W}_q, \mathbf{W}_k \in \mathbb{R}^{d \times d}$ are learnable matrices. Using these interaction scores, the node representation is updated as:

$$\hat{o}_i^{(l+1)}(t) = \hat{o}_i^{(l)}(t) + \sigma\left(\sum_{j^{t'} \in \mathcal{N}(i^t)} \alpha^{(l)}(i^t, j^{t'}) \mathbf{W}_v \tilde{o}_j^{(l)}(t')\right), \tag{30}$$

where $\mathcal{N}(i^t)$ denotes the neighbors of $i^t$, $\mathbf{W}_v \in \mathbb{R}^{d \times d}$ is a learnable value transformation matrix, and $\sigma$ represents a non-linear activation function. After propagating through $L$ layers, the observation representations are aggregated into

entity-level latent states. The final initial dynamic state of the observation is given by:

$$q_i(t) = \hat{o}_i^{(L)}(t) + TE(t), \quad h_i(t_0) = \frac{1}{M}\sum_{t=1}^{M}\sigma(\mathbf{W}_s q_i(t)), \tag{31}$$

where $M$ is the number of historical observations, and $\mathbf{W}_s \in \mathbb{R}^{d \times d}$ is a learnable transformation matrix. These latent states $\{h_i(t_0)\}_{i=1}^{N}$ serve as the initialized dynamic states for each agent. This approach transforms the enhanced dynamic representation $\hat{o}_i$ into temporally enriched and spatially aware representations, effectively capturing dependencies essential for modeling complex systems.

## E. Implement Details

The experiments are conducted using NVIDIA GeForce RTX 3090 GPUs as the hardware platform, coupled with Intel(R) Xeon(R) Gold 6240 CPU @ 2.60GHz. The deep learning framework employed was Pytorch, version 1.11.0, alongside CUDA version 11.3. We utilize the MLP for any projection models. The hidden layer size was set to 32 for each dataset. For optimization, the Adam optimizer (Kingma & Ba, 2014) was chosen, with a learning rate of $1e-5$ and a weight decay of $1e-3$ during the training process.

## F. Sensitivity (Q4)

To investigate the sensitivity of `GREAT` to the changes in substate $K$ and the dimension $p$, we analyze the performance across both ID and OOD settings. The analysis focuses on understanding how the model behaves when varying these key factors. As shown in Figure 7, we have inverted the RMSE values, so *higher* bars indicate *better* performance. We observe the following trends: **Obs. ❶** In any scenario, higher dimensions generally lead to better performance. Lower dimensions constrain the model's expressiveness, reducing the separability of projections, and thus fail to properly disentangle the dynamics. While increasing the dimension (e.g., to 64) brings some improvement, the effect becomes limited, and the computational overhead increases. Therefore, the choice of dimension should balance performance and computational efficiency. **Obs. ❷** In the ID setting, the number of substates has a relatively small impact on model performance. Smaller substate numbers are sufficient for simpler tasks. However, in the OOD setting, a higher number of substates contributes to a greater diversity of coupling patterns, which enhances the model's generalization ability. This aligns with the design motivation of our model, as higher substate numbers help capture more complex coupling relations, thus improving adaptability to unseen scenarios.

