# OpenReview forum: "Rethink GraphODE Generalization within Coupled Dynamical System"
_ICML.cc/2025/Conference — ICML 2025 spotlightposter_

### Official Review · Reviewer_CHTQ · 2025-03-11

**Overall Recommendation:** 5

**Summary:**

The paper investigates the generalization challenges of GraphODE models when used for coupled dynamical systems. It shows that mixing static attributes with dynamic states during initialization, along with an over-reliance on context-specific coupling patterns, can hurt the model’s performance in new settings. To overcome these issues, the paper introduces the GREAT framework, which features a Dynamic-Static Equilibrium Decoupler to clearly separate static and dynamic components, and a Causal Mediation for Coupled Dynamics module that uses variational inference to reduce misleading correlations. Experiments on systems like SPRING, CHARGED, and PENDULUM demonstrate that GREAT outperforms existing methods in both familiar and new environments.

## Update after rebuttal
After reviewing all the Reviewer-Authors discussions, I would like to express my strong support for this paper. Most reviewers have recognized the novelty and contributions of this work. The authors have presented an innovative framework that addresses fundamental challenges in GraphODE generalization for coupled dynamical systems.
I initially raised concerns about parameter sensitivity analysis and baseline method discussions. The authors' rebuttal thoroughly addressed these points with clear technical explanations and comprehensive empirical evidence. Their responses not only resolved my concerns but further strengthened my appreciation of their technical contributions.
Given the quality of both the original submission and their thorough rebuttal, I am raising my score.

**Claims And Evidence:**

All claims are well supported by evidence.

**Essential References Not Discussed:**

N/A

**Experimental Designs Or Analyses:**

Overall, the experimental setup and analyses clearly demonstrate the framework's enhanced performance and generalization capabilities.

**Methods And Evaluation Criteria:**

The paper uses RMSE and MAPE as key evaluation metrics. The proposed model and the evaluation criteria make sense for the problem.

**Other Comments Or Suggestions:**

Please see weaknesses.

**Other Strengths And Weaknesses:**

Strengths:

1)	Method is novel. The introduction of the DyStaED provides a novel mechanism to separate these features effectively, enabling better modeling of system evolution. The CMCD module employs variational inference for a nuanced mediation analysis of state evolution, offering deeper insights into how coupling dynamics influence outcomes.

2)	The paper innovatively combines causal inference with dynamic modeling to address the entanglement of static and dynamic components. It presents a systematic analysis using causal graphs, particularly in section 3.3, which details the state evolution and clarifies the influence of coupling dynamics. Theorem 3.1 is a standout, offering a rigorous theoretical foundation that significantly bolsters the framework’s claims.

3)	The paper is well-organized, with a clear progression from problem definition to theoretical analysis and experimental validation.

Weaknesses:

1）	Although the paper presents a solid experimental design, it would benefit from a more detailed sensitivity analysis regarding the choice of parameters, such as those used in variational inference or coupling patterns. This would help understand how sensitive the model is to variations in these parameters and further enhance its robustness.

2）	The paper provides limited discussion on baseline methods. A more detailed introduction of relevant baselines would offer clearer context for the improvements claimed.

**Questions For Authors:**

1.	How is variational inference implemented within the CMCD module? Is it integrated as part of the joint optimization during training, or is it applied as a separate post-processing step after feature extraction?

2.	Why does GREAT exhibit the lowest RMSE in both ID and OOD settings? Is there a specific characteristic of this method that makes it more robust in longer prediction lengths?

**Relation To Broader Scientific Literature:**

In terms of machine learning, the paper draws on work related to neural differential equations and graph neural networks. While these techniques have been applied successfully in other domains, their application to epidemic forecasting is a significant innovation. By leveraging these advanced techniques, the paper contributes to the growing body of research on using deep learning to enhance predictive modeling in epidemiology. This approach also connects to broader trends in the literature about the fusion of classical and modern machine learning techniques to address real-world problems.

**Theoretical Claims:**

Theoretical claims are correct and easy to understand.

---

> ### Author Rebuttal · Authors · 2025-04-01
>
> # Response to Reviewer CHTQ
>
> Thank you for your positive review and insightful questions. We are grateful for your recognition of our work's novelty and contributions. We address your comments below:
>
> > `Weakness 1`: Need more parameter sensitivity analysis.
>
> We appreciate this valuable feedback. For example, the orthogonality loss weight $\lambda_o$ is critical for balancing disentanglement and prediction accuracy. Our preliminary analysis shows:
>
> | Metric | $\lambda_o$=0.1 | $\lambda_o$=0.3 | $\lambda_o$=0.5 | $\lambda_o$=1.0 | $\lambda_o$=2.0 |
> |--------|--------------|-------------|------------|------------|-------------|
> | SPRING ID (RMSE) | 3.945 | 3.803 | **3.687** | 3.714 | 3.892 |
> | SPRING OOD (RMSE) | 3.986 | 3.754 | 3.651 | **3.619** | 3.785 |
>
> As shown above, $\lambda_o$=0.5 and $\lambda_o$=1.0 provide the optimal balance for ID and OOD settings respectively. Too small a value ($\lambda_o$<0.3) results in insufficient separation of static and dynamic components, while too large a value ($\lambda_o$>1.0) overly constrains the model's expressiveness. The model maintains strong performance across a reasonable range of values.
>
> > `Weakness 2`: Limited discussion of baseline methods.
>
> Thank for the friendly reminder. We will provide a more comprehensive discussion of baseline methods in the revised version.
>
>
> > `Question 1`: Implementation details of variational inference?
>
> Our CMCD module implements variational inference by treating coupling factors $\eta_i(t)$ as confounding variables, parameterized via $q(\eta_i(t)|h_i(t))$ with $\zeta_i(t) = \text{Softmax}(W_\eta h_i(t))$. We employ Gumbel-Softmax sampling for differentiability: $\eta_{i,k}(t) = \frac{\exp((\zeta_{i,k}(t)+g_k)/\tau)}{\sum_j \exp((\zeta_{i,j}(t)+g_j)/\tau)}$. The KL divergence term $\text{KL}(q(\eta_i(t)|h_i(t))\|p(\eta_i(t)))$ regularizes against a context-agnostic prior, reducing spurious correlations. This approximates the interventional likelihood $\log p(y_i(T)|\text{do}(h_i(t_0)),\mathcal{G})$, enabling our model to learn universal physical dynamics rather than domain-specific patterns. All components are jointly optimized, allowing the coupling inference to leverage both static and dynamic information for robust generalization.
>
>
> > `Question 2`: Reason for superior performance in longer predictions?
>
> GREAT's superior long-term prediction performance stems from several key mechanisms working in concert: First, our effective disentanglement prevents error accumulation by maintaining consistent static components while allowing dynamic components to evolve naturally. Second, the CMCD module enables coupling-aware evolution that explicitly models inter-node influences, capturing complex interdependencies crucial for extended horizons. Third, the DHPA component captures self-exciting temporal patterns at multiple scales, effectively modeling both short-term fluctuations and long-term trends. Finally, empirical stability analysis shows our error growth rate is significantly lower in Figure 6. These advantages make GREAT particularly suitable for applications requiring extended forecasting horizons in complex physical systems.

---

> > ### Comment · Reviewer_CHTQ · 2025-04-02
> >
> > The authors have solved my concerns. Thus, I vote for the acceptance of this paper.

---

> > > ### Author Response · Authors · 2025-04-02
> > >
> > > Dear reviewer CHTQ
> > >
> > >
> > > We sincerely appreciate your thoughtful feedback and your willingness to consider the merits of our work.
> > >
> > > Best regards,
> > >
> > > Authors

---

### Official Review · Reviewer_XMqH · 2025-03-13

**Overall Recommendation:** 5

**Summary:**

This paper presents the GREAT framework to improve GraphODE models' generalization in coupled dynamical systems. It tackles two key issues: the entanglement of static and dynamic information during initialization and the reliance on environment-specific coupling patterns. The framework introduces two modules: the DyStaED for separating features via orthogonal projections, and CMCD, which uses variational inference to disentangle latent coupling factors and reduce spurious correlations. Validated through extensive experiments, the approach shows significant improvements in both in-distribution and out-of-distribution scenarios.

**Claims And Evidence:**

Based on the description, the claims in the paper appear to be well-supported by evidence. The paper does an excellent job in aligning its problem description with the motivation provided by the SCM causal analysis. It accurately and intuitively identifies the issues in existing methods.

**Essential References Not Discussed:**

All related references have been included.

**Experimental Designs Or Analyses:**

The experimental design is robust and comprehensive, incorporating both ID and OOD evaluations across simulated systems like SPRING, CHARGED, and PENDULUM. The comprehensive ablation studies (Figure 5) effectively isolate and highlight the contributions of both the DyStaED and the CMCD modules.

**Methods And Evaluation Criteria:**

The paper uses RMSE and MAPE as key evaluation metrics. RMSE captures overall accuracy and penalizes larger errors, while MAPE provides a clear percentage-based evaluation, making it ideal for comparing relative errors. The use of both ID and OOD scenarios is also robust. ID testing ensures good performance in familiar conditions, while OOD evaluation tests the model’s ability to generalize in new, unseen environments.

**Other Comments Or Suggestions:**

The author should provide clearer details on the baseline in the appendix—specifically, parameter settings, configurations, or any implementation differences.

**Other Strengths And Weaknesses:**

### Paper Strength

-	The GREAT framework rethinks traditional approaches by decoupling static and dynamic attributes using causal inference, which allows the model to effectively mitigate confounding factors and achieve superior generalization.
-	The paper establishes a robust theoretical framework by integrating causal inference into the coupling patterns within GraphODE models, while its extensive experiments—spanning multiple simulated datasets and testing both ID and OOD conditions—thoroughly validate the proposed approach.
-	The paper features a clearly articulated problem description and a compelling motivation grounded in Structural Causal Models (SCM), which together provide a great foundation for the proposed method.
-	The paper’s diagrams and figures are exceptionally well-designed, combining aesthetic appeal with clarity to effectively convey complex concepts and experimental results.


###  Paper Weakness

-	Although the paper acknowledges some inherent constraints, it does not thoroughly explore the potential limitations of its approach.

**Questions For Authors:**

1.	In scenarios where the distinction between static and dynamic features is ambiguous, how robust is the decoupling mechanism, and what measures are in place to prevent misclassification?
2.	Are there experiments to verify the stability of the GREAT model under different training rounds and initial conditions? If the model exhibits instability or significant fluctuations in certain situations, can it be interpreted as limitations in some assumptions or parameter choices of the model?

**Relation To Broader Scientific Literature:**

The paper leverages advanced AI techniques to deepen our understanding of physical systems. It builds on earlier work that applied machine learning to nonlinear dynamics, offering fresh interdisciplinary insights that expand the boundaries of AI in physics.

**Theoretical Claims:**

The paper provides a strong, systematic causal analysis using causal graphs to identify and explain the key issues in existing methods. The causal analysis is thorough and offers an insightful understanding of the challenges faced by GraphODE models, leading to constructive Generalizable GraphODE Design Principle. Furthermore, Theorem 3.1 is a key contribution, providing a clear and convincing theoretical result that underpins the effectiveness of the proposed causal mediation approach. The theorem articulates a critical relationship between state evolution and coupling dynamics, offering a rigorous justification for the model's ability to generalize across both ID and OOD scenarios.

---

> ### Author Rebuttal · Authors · 2025-04-01
>
> # Response to Reviewer XMqH
>
> Thank you for your thorough review and encouraging feedback. We are grateful for your positive assessment about novelty and importance of our work. We address your questions below:
>
> > `Weakness`: Limited exploration of approach limitations.
>
> We fully agree that a more thorough discussion of limitations would significantly strengthen the paper. In the revised version, we will add a dedicated limitations section that provides comprehensive analysis. One potential limitation is training stability: Similar to many deep learning approaches utilizing variational inference, our method requires careful tuning to balance multiple loss components (prediction loss, orthogonality loss, and KL divergence term). We will provide detailed discussion of these potential training challenges along with practical strategies for mitigating them through hyperparameter selection and optimization techniques.
>
>
> > `Comment`: Need clearer baseline implementation details.
>
> We sincerely thank you for this suggestion. We will enhance the baseline description and implementation details in the revised version.
>
> > `Question 1`: Robustness when static/dynamic distinction is ambiguous?
>
> This is an excellent question. Our decoupling mechanism employs several strategies to maintain robustness:
> 1) Orthogonality regularization ($L_o$): The loss function in Eq.8 enforces strict orthogonality between the static and dynamic subspaces, helping to separate features even when they appear entangled. Our experiments show that without this regularization, performance drops by 12.3% on average in OOD settings.
> 2) Learnable subspaces: Rather than using predetermined feature splits, our approach learns the separation through end-to-end training. The embedding layers $S_{static}$ and $S_{dynamic}$ adaptively determine the optimal projection for each domain, allowing the model to discover natural separations even in ambiguous cases.
> 3) We employ a synergy decoder (Eq.12) that allows information to flow from both static and dynamic components. This ensures that even if some features are misclassified during training, critical information is not lost during reconstruction.
> 4) We introduce a DHPA that dynamically adjusts the importance of different feature patterns at multiple scales. This helps the model better handle ambiguous cases by learning to focus on the most relevant patterns for dynamic separation at each level of abstraction. Our experiments show this further improves robustness in Figure 5.
>
>
> > `Question 2`: Stability across different training configurations?
>
> We have conducted extensive experiments to verify GREAT's stability across different training configurations. The final reported results from all tables are the result of 5 random seeds.
> Here, we also tested various learning rates and training epochs on the SPRING dataset. The results demonstrate that GREAT maintains stable performance across a range of configurations:
>
> | Metric | LR=1e-3 | LR=5e-4 | LR=1e-4 | LR=5e-5 | LR=1e-5 |
> |--------|---------|---------|---------|---------|---------|
> | SPRING ID (RMSE) | 3.854 | 3.792 | 3.736 | 3.705 | **3.687** |
> | SPRING OOD (RMSE) | 3.814 | 3.769 | 3.726 | 3.683 | **3.619** |
>
> | Epochs | 100 | 200 | 300 | 400 | 500 |
> |--------|-----|-----|-----|-----|-----|
> | SPRING ID (RMSE) | 4.243 | 3.986 | 3.782 | **3.687** | 3.721 |
> | SPRING OOD (RMSE) | 4.279 | 3.947 | 3.751 | **3.619** | 3.655 |

---

> > ### Comment · Reviewer_XMqH · 2025-04-02
> >
> > Thanks for author detailed response. After reading the rebutal and other reviewer questions, most of my concerns have been addressed. I'm happy to increase my score.

---

> > > ### Author Response · Authors · 2025-04-02
> > >
> > > Dear Reviewer XMqH,
> > >
> > > Thank you again for recognizing the importance of our work.
> > >
> > > Best regards,
> > >
> > > Authors

---

### Official Review · Reviewer_qXoi · 2025-03-13

**Overall Recommendation:** 3

**Summary:**

The paper proposes a new graphODE-based method to process the compiled dynamical system. The trajectory is decomposed into dynamic and static parts in the latent space. The experiments demonstrate the effectiveness of the proposed method.

**Claims And Evidence:**

Yes

**Essential References Not Discussed:**

None

**Experimental Designs Or Analyses:**

I'm not sure the OOD split according to the system parameters makes sense.

**Methods And Evaluation Criteria:**

Yes

**Other Comments Or Suggestions:**

-  It is better to visualize the orthogonality loss L_o, which would be helpful to let the reader know if the decomposition is successful.
- The test sets seem all about the small particle system; those particles don't have strong interaction with each other. Can the method test on molecular systems, such as MD17 and QM9 in [1].

[1] Geometric Trajectory Diffusion Models, NIPS 2024

**Other Strengths And Weaknesses:**

Strengths:
- The paper is well-written and well organized.
- The idea of decomposing the trajectory to static and dynamic parts in the latent space is interesting and technique sound.
- This paper also provides many useful insights, detailed discussions, and theoretical support.

Weakness:
- The introduction of background on coupling dynamic is limited.  A detailed background on coupled dynamical systems contributes to a better understanding.
- The experiments only contain 3 small datasets. More experiments on other domains like molecule or protein will make the paper more convincing.

**Questions For Authors:**

None

**Relation To Broader Scientific Literature:**

GraphODE is a previous work.

**Theoretical Claims:**

No

---

> ### Author Rebuttal · Authors · 2025-04-01
>
> # Response to Reviewer qXoi
>
> Thank you for your constructive comments and positive assessment of our work. We address your concerns below and hope our responses will help update your score:
>
> > `Weakness 1`: Limited background on coupling dynamics.
>
> We sincerely appreciate this valuable suggestion. Due to space limitations, we provided only essential background information. We will expand the background section in the revised version to include more examples and fundamental principles of coupled dynamical systems.
>
> > `Weakness 2 & Comment 2`: Small datasets and need molecular system testing.
>
> We greatly appreciate these insightful suggestions. We respectfully would like to clarify several points regarding the systems in our experiments:
>
> 1) While appearing simple, our systems exhibit strong interaction patterns: the CHARGED system demonstrates complex electromagnetic interactions with high coupling strengths, and the PENDULUM system captures chaotic behaviors with highly entangled dynamics.
>
> 2) Following standard settings in previous works [1,2], these systems were deliberately chosen as they present the core challenges (static-dynamic entanglement and coupling bias) that our method addresses. These **challenges are fundamental** across coupled systems regardless of complexity level and have been widely recognized in the field.
>
> 3) Our OOD experiments already demonstrate generalization across varying interaction strengths by testing with different coupling parameters (e.g., interaction strength γ∈[0.05,0.15] for SPRING and γ∈[0.5,1.5] for CHARGED in OOD settings).
>
> 4) We appreciate your suggestion about molecular systems and have conducted preliminary experiments on MD17:
>
> | Method     | Aspirin | Benzene | Ethanol | Malonaldehyde | Naphthalene | Salicylic | Toluene | Uracil |
> |------------|---------|---------|---------|---------------|-------------|-----------|---------|--------|
> | LatentODE  | 0.667   | 0.809   | 0.272   | 0.698         | 0.132       | 0.186     | 0.351   | 0.154   |
> | LG-ODE     | 0.436   | 0.581   | 0.240   | 0.348         | **0.046**   | 0.092     | 0.157   | 0.091   |
> | PGODE      | 0.398   | 0.534   | 0.163   | 0.325         | 0.057       | 0.085     | 0.134   | 0.074   |
> | GREAT      | **0.252** | **0.123** | **0.156** | **0.317**  | 0.048     | **0.079**  | **0.127** | **0.064** |
>
> We report RMSE values (Å) on MD17 dataset with irregular temporal sampling where only 60% of timesteps were randomly sampled and follow other stanard setting in our manuscript. GREAT also performs well on molecular systems because these systems naturally exhibit the same challenges our method addresses: varying interaction patterns. Our approach effectively captures both local and global dynamics, achieving the best performance on 7 molecules.
> We acknowledge the value of testing on diverse systems and will include more comprehensive results including QM9 in the revised version.
>
>
>
> > `Comment 1`: Need visualization of orthogonality loss.
>
> Thank you for this valuable suggestion. The orthogonality loss trajectory during training on SPRING:
>
> | Epoch: | 1 | 50 | 100 | 150 | 200 | 300 | 400 |
> |--------|---|----|----|----|----|----|----|
> | Loss (Lo): | 13.88 | 5.42 | 2.11 | 1.25 | 0.67 | 0.55 | 0.54 |
>
> As shown, the orthogonality loss steadily decreases during training, indicating effective disentanglement of static and dynamic components. We have also demonstrated its effectiveness through ablation studies in Sec. 4.4 and Figure 5. We will include more detailed visualizations in the revised version.
>
> > `Concern 1 (Experimental Designs Or Analyses)`: OOD setting.
>
> Our OOD split based on system parameters follows established practices in prior work [1] and directly tests the primary challenge in coupled dynamical systems: generalization across varying initialization conditions. This design has strong physical meaning as parameter changes, fundamentally alter system dynamics.
>
>
>
>
> [1]: Pgode: Towards high-quality system dynamics modeling. In ICML, 2024.
> [2]: Physics-informed regularization for domain-agnostic dynamical system modeling. In NeurIPS, 2024.

---

> > ### Comment · Reviewer_qXoi · 2025-04-02
> >
> > Thanks for the response！ The replies address my concern. I maintain my score.

---

> > > ### Author Response · Authors · 2025-04-02
> > >
> > > Dear Reviewer qXoi,
> > >
> > > Thank you again for recognizing the innovation and contribution of our work and for your willingness to support its acceptance!
> > >
> > > Best regards,
> > >
> > > Authors

---

### Official Review · Reviewer_LVHe · 2025-03-15

**Overall Recommendation:** 2

**Summary:**

This paper introduces a new framework named GREAT (Generalizable GraphODE with disEntanglement And regularizaTion) to enhance the generalization capabilities of Graph Ordinary Differential Equations (GraphODE) models for coupled dynamical systems. The key contributions and findings include identifying generalization challenges in GraphODE and devise corresponding modules to overcome. The results in three datasets show good performance in both in-distribution (ID) and out-of-distribution (OOD) settings.

**Claims And Evidence:**

The two main challenges claimed in this paper are appropriately tackled in the method design.

However, I am not fully convinced by the presented results since only average performance metrics are included. For example,

- For challenge 1, does the proposed method correctly decouple static and dynamic parts?

- For challenge 2, does the proposed method correctly capture the coupling factor?

**Essential References Not Discussed:**

Enough.

**Experimental Designs Or Analyses:**

I think more ablation studies and in-depth analyses are needed to demonstrate that the proposed method successfully overcomes two challenges.

**Methods And Evaluation Criteria:**

Most parts of the method design are reasonable. However,

- The motivation behind Dynamic Hawkes Process Augmentation is unclear. Is it universal to all kinds of dynamical systems?

**Other Comments Or Suggestions:**

Figure 3 is Beautiful, but I think the icons are too much, and some of them are not necessary.

**Other Strengths And Weaknesses:**

Strength

- Rigorous deriviation and method design.

- Well-motivated in terms of two main challenges.

- Good performance compared with several SOTA methods.

Weakness (Detailed in above comments)

- The covered systems seem too simple. For example, the dimensionality of these systems is no greater than 5.

- Lack of motivation for Dynamic Hawkes Process Augmentation.

- Performance evaluations cannot fully validate the claimed advantages.

**Questions For Authors:**

Please see my comments above.

**Relation To Broader Scientific Literature:**

Coupled dynamic systems are ubiquitous across domains. However, the conducted experiments only cover a small range of simplified systems.

**Theoretical Claims:**

Checked.

---

> ### Author Rebuttal · Authors · 2025-04-01
>
> # Response to Reviewer LVHe
> Thank you for your thorough review and constructive feedback. We address your concerns below and hope these clarifications will help you re-evaluate and update the score:
>
> > `Weakness 1`: Systems too simple with low dimensionality (≤5).
>
> Thank you for this observation. We would like to offer several considerations that guided our experimental design:
>
> 1) These systems exhibit complex nonlinear dynamics and chaotic behaviors. We calculated the Maximum Lyapunov Exponent (MLE) for each system, where positive values indicate chaotic dynamics:
>
>     |SPRING|CHARGED|PENDULUM|
>     |-|-|-|
>     |0.651|0.875|18.932|
>
>     As shown, all systems exhibit positive MLEs, with PENDULUM showing strong chaos (MLE≫0). The SPRING system captures elastic interactions with long-range dependencies, the CHARGED system models electromagnetic forces with phase transitions, and the PENDULUM system demonstrates extreme sensitivity to initial conditions—each representing distinct coupling mechanisms common in real-world applications.
>
> 2) Our research addresses two challenges in GraphODE generalization: static-dynamic entanglement and coupling pattern bias. These challenges **are intrinsic to coupled systems regardless of dimensionality**.
>
> 3) Following established practice [1,2], these systems serve as standard benchmarks enabling fair comparison with prior work.
>
> 4) We conducted additional experiments on a higher-dimensional SPRING system (10 particles) and the MD17 molecular dynamics (please refer to response to Reviewer qXoi), where GREAT consistently outperformed baselines:
>
>     |RMSE|LatentODE|LG-ODE|PGODE|GREAT|
>     |-|-|-|-|-|
>     |ID|7.65|6.27|4.92|4.29|
>     |OOD|8.37|6.81|5.39|5.01|
>
> [1]: Neural Relational Inference for Interacting Systems. In ICML, 2018.
> [2]: Physics-informed regularization for domain-agnostic dynamical system modeling. In NeurIPS, 2024.
>
> > `Weakness 2`: Unclear motivation for Dynamic Hawkes Process Augmentation.
>
> DHPA addresses a fundamental challenge: coupled systems exhibit complex temporal dependencies that basic representations cannot capture. While our disentanglement separates static and dynamic features, the dynamic component needs enhancement to model multi-scale patterns. Following a "divide and conquer" strategy, DHPA strengthens the dynamic representation with weighted historical information: $ \hat{o}\_i(t) = o_i(t) + \delta\sum_{\tau=1}^s w_{\tau,t}\cdot o_i(t-\tau) $. This approach is **universal across systems** because its adaptive weights $ w_{\tau,t} $ automatically adjust to each system's temporal characteristics, with its effectiveness confirmed by our ablation studies (Fig.5).
>
> > `Weakness 3`: Insufficient validation of claimed advantages.
>
> We provided ablation studies in Figure 5 and Section 4.4 demonstrating each component's contribution. Additional evidence validates our key contributions:
>
> Challenge 1: Static-Dynamic Disentanglement: The orthogonality loss decreases during training, showing successful separation:
>
> |Epoch|1|100|200|400|
> |-|-|-|-|-|
> |SPRING|13.88|2.11|0.67|0.54|
> |CHARGED|18.94|5.63|2.15|1.62|
> |PENDULUM|8.76|1.58|0.63|0.64|
>
> Feature separation testing: Mutual information (MI) analysis between static and dynamic representations with permutation tests: https://anonymous.4open.science/r/Response_to_Reviewer_LVHe-F29E/table-MI.md. MI was estimated using the Kraskov estimator across 1000 time points, with permutation tests (10,000 shuffles). Decreasing p-values confirm statistically significant separation. The steady MI decrease demonstrates our model eliminates information leakage between representation spaces.
>
> Temporal consistency: Cosine similarity between representations at t=0 and later points confirms static representations maintain higher consistency: https://anonymous.4open.science/r/Response_to_Reviewer_LVHe-F29E/table-time-consistency.md
>
> Challenge 2: Coupling Pattern Capture: Our experiments and ablation studies demonstrate GREAT's superior performance in OOD scenarios. Our theoretical analysis in Section 3.3 provides rigorous justification by formulating the problem through a causal lens - modeling coupling factors as confounding variables and deriving an interventional likelihood $p(y_i(T)|do(h_i(t_0)), \mathcal{G})$ rather than observational likelihood. Theorem 3.1 proves this formulation captures intrinsic physical dynamics invariant, going beyond empirical validation to provide theoretical guarantees.
>
> We will include more detailed analyses in the revised version.
>
> > `Other Comment`: Excessive icons in Figure 3.
>
> We agree and will simplify Figure 3 while maintaining the figure's informational content.

---

### Decision · Program_Chairs · 2025-05-01

**Decision:**

Accept (spotlight poster)

**Comment:**

This paper aims to improve the generalization of Graph Ordinary Differential Equations (GraphODEs) in the case of coupled dynamical systems. The authors use a Structural Causal Model to examine current GrapheODEs and design mitigation modules in the form of two systems:  Dynamic-Static Equilibrium Decoupler (DyStaED) and Causal Mediation for Coupled Dynamics (CMCD). The former focuses on the disentangling of static and dynamic states at the initialization step, while the latter estimates latent causal factors through variational inference. The authors show that these methods can improve in- and out- of distribution performance and apply the method to a number of datasets.

While the reviewers noted a few shortcomings, such as the potential simplicity of some of the examples, the overall interactions with the authors were able to mitigate most of these concerns and so I recommend this work be accepted.